# Nanoparticle-Enabled Combination Therapy Showed Superior Activity against Multi-Drug Resistant Bacterial Pathogens in Comparison to Free Drugs

**DOI:** 10.3390/nano12132179

**Published:** 2022-06-24

**Authors:** Amarpreet Brar, Satwik Majumder, Maria Zardon Navarro, Marie-Odile Benoit-Biancamano, Jennifer Ronholm, Saji George

**Affiliations:** 1Department of Food Science and Agricultural Chemistry, Macdonald Campus, McGill University, 21111 Lakeshore, Ste Anne de Bellevue, QC H9X 3V9, Canada; amarpreet.brar@mail.mcgill.ca (A.B.); satwik.majumder@mail.mcgill.ca (S.M.); jennifer.ronholm@mcgill.ca (J.R.); 2Swine and Poultry Infectious Diseases Research Center (CRIPA), Faculté de Médecine Vétérinaire, Université de Montréal, 3200 Sicotte, Saint-Hyacinthe, QC J2S 2M2, Canada; maria.zardon@umontreal.ca (M.Z.N.); marie-odile.benoit-biancamano@umontreal.ca (M.-O.B.-B.); 3Research Group on Infectious Diseases in Production Animals (GREMIP), Department of Pathology and Microbiology, Faculté de Médecine Vétérinaire, Université de Montréal, 3200 Sicotte, Saint-Hyacinthe, QC J2S 2M2, Canada; 4Department of Animal Science, Macdonald Campus, McGill University, 2111 Lakeshore, Ste Anne de Bellevue, QC H9X 3V9, Canada

**Keywords:** multi-drug resistance, combination therapy, nanoparticle, antimicrobial resistance, *Staphylococcus*, *Salmonella*, animal agriculture, pathogen

## Abstract

The emergence of multidrug-resistant (MDR) bacterial pathogens in farm animals and their zoonotic spread is a concern to both animal agriculture and public health. Apart from antimicrobial resistance (AMR), bacterial pathogens from the genera of *Salmonella* and *Staphylococcus* take refuge inside host cells, thereby demanding intervention strategies that can eliminate intracellular MDR pathogens. In this study, seven clinical isolates of *Salmonella* and *Staphylococcus* from swine farms were characterized for antibiotic (n = 24) resistance, resistance mechanisms, and virulence characteristics. All isolates showed resistance to one or more antibiotics and *S. enterica* ser. Typhimurium isolate had the highest resistance to the panel of antibiotics tested. Major resistance mechanisms identified were efflux pump and beta-lactamase enzyme activities. *Staphylococcus* isolates showed complete hemolysis and strong biofilm formation, while *Salmonella* isolates caused partial hemolysis, but showed no or weak biofilm formation. MDR isolates of *S. aureus* M12 and *S. enterica* ser. Typhimurium bacteria were subsequently tested against combinations of antibiotics and potentiating adjuvants for improved antibacterial efficacy using a checkerboard assay, and their fractional inhibitory concentration index (FICI) was calculated. A combination of chitosan and silica nanoparticles containing tetracycline (TET) and efflux pump inhibitor chlorpromazine (CPZ), respectively, was characterized for physicochemical properties and effectiveness against MDR *Salmonella enterica* ser. Typhimurium isolate. This combination of nano-encapsulated drugs improved the antibacterial efficacy by inhibiting AMR mechanisms (efflux activity, beta-lactamase enzyme activity, and hydrogen sulfide (H_2_S) production) and reducing intracellular pathogen load by 83.02 ± 14.35%. In conclusion, this study sheds light on the promising applicability of nanoparticle-enabled combination therapy to combat multidrug-resistant pathogens encountered in animal agriculture.

## 1. Introduction

The prevalence of pathogenic microorganisms in food animals and their possible zoonotic spread through direct contact or the food chain pose public health concerns [1,2]. Common bacterial pathogens infecting farmed animals can cause various infections, often requiring treatment with antibiotics. In addition, antibiotics are used as prophylactics and as antimicrobial growth promoters (AGPs) although the farm use of antimicrobials reserved for infection control in humans has been restricted in the recent past in countries such as Canada [3,4]. Nonetheless, the use of antibiotics in animals is an important driver of the emergence and spread of antimicrobial-resistant (AMR) bacterial human infections [1,5,6]. AMR pathogens in animals and their zoonotic potential necessitates the development of alternate therapeutics for disease management of multi-drug resistant (MDR) pathogens [7]. 

Combining antibiotics with potentiating adjuvants is one such solution that can enhance the potency of antibiotics and increase the sensitization of bacteria to commonly used antibiotics. For instance, adjuvants such as efflux pump inhibitors have shown to subdue the resistance mechanisms of bacterial cells and promote the action of antibiotics and silver [8,9]. Combination therapy—a treatment modality of using a combination of drugs is successful in boosting the effectiveness of antibiotics [10,11]. However, combination therapy with free forms of drugs lacks target specificity, control over release kinetics, and has a shorter half-life in the body [12,13]. These drawbacks highlight the urgent need to devise alternate modes of drug delivery systems.

Recent developments in nanoparticle-based drug delivery have provided numerous promising opportunities in terms of longer circulation time, controlled drug release, ability to target specific tissue sites, and to deliver more than one drug at the same time [14,15,16]. In addition, nanoparticle-based drug delivery systems can be custom-designed to meet desired properties. For instance, chitosan nanoparticles are mucoadhesive, biodegradable, positively charged, and easily surface-modifiable to target specific sites inside the body, thus making chitosan nanoparticles (Ch NPs) a promising candidate for drug delivery [17]. Silica nanoparticles (Si NPs) have complimentary advantageous features such as high drug loading capacity, biocompatibility, and surface modification possibilities [18,19].

In this study, *Staphylococcus* and *Salmonella* isolates from Quebec-based animal farms were studied for AMR and major AMR mechanisms such as efflux pump activity and beta-lactamase enzyme activity were identified. In addition, virulence characteristics such as biofilm formation, and hemolysis relevant for bacterial survival inside the host body were also investigated. The effectiveness of combining antibiotics with adjuvant molecules was tested in vitro to identify optimal combinations for neutralizing MDR strains. We further showed that delivering identified combinations of drug molecules using nanoparticles could enhance antibacterial efficiency against MDR strains evading human epithelial cells.

## 2. Materials and Methods

### 2.1. Isolates and Chemicals

Seven pathogenic bacterial isolates from Quebec farm animals, submitted for diagnostic purposes to the Complexe de diagnostic et d’épidémiosurveillance vétérinaires du Québec (CDEVQ), Saint-Hyacinthe, QC, Canada, with diseases such as exudative epidermitis, diarrhea, or septicemia, were selected for this study. These isolates consisted of *S. enterica* serovar Choleraesuis, *S. enterica* serovar Dublin, *S. enterica* serovar Typhimurium, *Staphylococcus aureus* M12, *S. aureus* M17, *S. hyicus* M43, and *S. hyicus* M48. *S. aureus* ATCC 25923 and *E. coli* ATCC 25922 (Oxoid company, Nepean, ON, Canada) were used as quality control (QC) strains. The sodium salt of resazurin, Silica, mesostructured HMS (wormhole) (Catalog number 541036-5G) with pore size 2–4 nm, and ethidium bromide (EtBr) were purchased from Sigma-Aldrich, Canada. Chitosan (85% deacylation) was obtained from Alfa Aesar (Thermo Fisher Scientific, Tewksbury, MA, USA). Tetracycline (TET) and chlorpromazine (CPZ) were purchased from Bio Basic, ON, Canada. Nitrocefin dye was purchased from Abcam, Cambridge, United Kingdom.

### 2.2. Antimicrobial Susceptibility Determination

The susceptibility of bacterial isolates to 24 antibiotics (Oxoid, Thermo Fischer Scientific, Nepean, ON, Canada) (Appendix A) was determined by the Kirby–Bauer disk diffusion method according to Clinical and Laboratory Standards Institutes Guidelines [20,21], and QC strains were used as controls.

### 2.3. Testing Combinations of Antibiotics and Adjuvants

Antibiotics of ampicillin, chloramphenicol, TET, and trimethoprim were tested for additive or synergistic effects, as they are commonly used in food animals and humans [22,23]. The selected adjuvants consisted of an efflux pump inhibitor (EPI) (Chlorpromazine-CPZ), a beta-lactamase inhibitor (tazobactam), and essential oils (EOs) (thymol EO, cinnamon EO, and oregano EO) with different modes of action on bacterial cells. The composition of EOs used in this study was reported previously [24].

MICs of the antibiotics and adjuvants were determined against *S. enterica* ser. Typhimurium and *S. aureus* M12 using the microtiter broth dilution method. A checkerboard assay was used to categorize the type of antimicrobial effect of the tested combinations [25]. Antibiotics were diluted column-wise, and the adjuvants were diluted row-wise in a 96-well plate. Each well contained a unique concentration of drug and adjuvant to which 10 µL of cells (0.5 McFarland) were added. The 96-well plates were incubated at 37 °C overnight. Cell viability was checked using a resazurin assay (Appendix A) [26]. The well with no growth of bacteria (baseline reading for fluorescence) was considered for determining the MIC. The fractional inhibitory concentration index (FICI) was calculated according to the following formula.
(1)$${\mathrm{FIC}}_{A}=\frac{A}{{\mathrm{MIC}}_{A}};\;{\mathrm{FIC}}_{B}=\frac{B}{{\mathrm{MIC}}_{B}}$$
FICI = FIC_A_ + FIC_B_(2)
where A and B are the MICs of components A and B in combination, respectively. FICI values FICI ≤ 0.5, 0.5 < FICI ≤ 1, 1 < FICI ≤ 4, and FICI > 4 were categorized as synergistic, additive, indifferent and antagonistic, respectively.

### 2.4. Synthesis of Drug-Loaded Nanoparticles and Combination Drug (CMD)

TET-loaded Ch NPs (Ch-TET NPs) were synthesized using the ionic gelation method using TPP, as reported earlier [27]. Accordingly, chitosan (50 mg) was dissolved in 25 mL of 1% (*v*/*v*) acetic acid and stirred for 48 h. The pH was adjusted to 4.5–4.6 using 1.0 N NaOH and the suspension was sonicated for 10 min. Subsequently, 3.4 mL of the 1% (*w*/*v*) TET solution was added under constant stirring. TPP was added dropwise to the resulting solution where the chitosan:TPP ratio was 3:1. The resulting suspension was sonicated for 10 min and stirred for another hour to obtain Ch-TET NPs. The resulting NPs were collected by centrifuging the suspension at 11,000× *g* for 12 min and discarding the supernatant. Sedimented particles were washed twice with de-ionized (DI) water and were then resuspended in DI water.

For the synthesis of CPZ-loaded Si NPs (Si-CPZ NPs), 40 mg of mesoporous Si NPs were dispersed in 10 mL of DI water. CPZ was dissolved in DI water to obtain 1% (*w*/*v*) solution, 4 mL of which was added dropwise to the Si NPs suspension. The suspension mixture was stirred for 24 h, and then freeze-dried to obtain a white powder containing Si-CPZ NPs. Finally, CMD was obtained by combining Ch-TET NPs and Si-CPZ NPs in a 1:1 ratio in DI water.

### 2.5. Determination of Size and Surface Charge

Dynamic light scattering (DLS) (NanoBrook Omni instrument, Brookhaven’s, Holtsville, NY, USA) was used to measure the surface charge and hydrodynamic size of particles dispersed in DI water (50 µg/mL). The particle suspension was loaded into a pre-rinsed folded capillary cell and a voltage of 100 V was applied for the measurement of zeta potential.

### 2.6. Scanning Electron Microscopy (SEM) Observation

Particle morphology was examined by SEM (FEI Quanta 450 Field Emission Scanning Electron Microscope, Phillips, Austin, TX, USA). Nanoparticle samples were prepared by serial dehydration in increasing concentrations of absolute ethanol in DI water (30%, 50%, 70%, and 90%) for 10 min each and resuspended in 100% ethanol at a concentration of 50 µg/mL. A small amount (10 µL) of the sample was dropped onto a copper grid and allowed to dry at room temperature. These particles were then coated with platinum before acquiring SEM images.

### 2.7. FTIR Spectra of Synthesized Particles

Attenuated total reflectance-Fourier transform infrared (ATR-FTIR) (ALPHA-P, Bruker, Billerica, MA, USA) was used to analyze the surface functional groups of the particles [28]. For this, 5 µL of the nanoparticle suspension was dropped on the ATR probe and allowed to dry. FTIR spectra were obtained using a wavelength range of 600–4000 cm^−1^, with a resolution of 4 cm^−1^, and 32 scans. The obtained data were analyzed using the OMNIC 8.2.0.387 software.

### 2.8. Determination of Encapsulation Efficiency (EE)

Antibiotic-loaded-nanoparticles were dispersed in DI water (1 mg/mL). The suspensions were sonicated for 30 min, followed by centrifugation at 40,000× *g* for 40 min to remove loaded antibiotics from respective nanoparticle carriers. The supernatant was used to quantify the amount of TET and CPZ loaded by measuring the absorbance using a spectrophotometer (Genessys^TM^ 10S, Thermo Fisher, Tewksbury, MA, USA) at 285 nm and 305 nm wavelengths, respectively [29]. The total amount of antibiotic-loaded nanoparticles was calculated from a standard curve prepared using known concentrations of TET and CPZ.

### 2.9. Determination of Release Behavior

The release behavior of loaded nanoparticles was studied by dispersing nano-encapsulated drug samples in PBS at pH 4.3 (representing the gastric pH) and at pH 6.3 (representing the intestinal pH of swine). The samples were centrifuged at 2000× *g* for 3 min and the supernatant was removed every 24 h for seven days. The absorbance (285 nm and 305 nm, respectively, for TET and CPZ) of the supernatant was measured as a function of time using a spectrophotometer (Genessys^TM^ 10S, Thermo Fisher, Tewksbury, MA, USA) to determine the time bound release of TET and CPZ released.

### 2.10. Antibacterial Activity of Nano-Encapsulated Antibiotic and/or Efflux Pump Inhibitor

Antibacterial efficiency of Ch-TET NPs, Si-CPZ NPs, and CMD was determined against MDR bacteria- *S. enterica* ser. Typhimurium and *S. aureus* M12. Test compounds/particles were subjected to twofold serial dilution in a sterile 96-well plate. Bacterial cultures (10 µL) adjusted to 0.5 McFarland standard (obtained from Mueller–Hinton Broth (MHB) inoculated with a single colony of bacteria and incubated overnight at 37 °C) were added to the wells. The plates were incubated at 37 °C for 18 h under mild shaking. Subsequently, bacterial viability was monitored using a resazurin assay. *S. aureus* ATCC 25923 and *E. coli* ATCC 25922 were used as quality control strains.

## 3. Effect of Nano-Encapsulated Drugs on Bacterial Resistance Mechanisms

### 3.1. Beta-Lactamase Inhibition Assay

The beta-lactamase inhibition was tested as mentioned previously [7]. Briefly, the bacterial isolates (100 µL at 1.0 McFarland standard) were exposed to sub-lethal concentrations of test compounds (MIC × $\;\frac{1}{3}$), and were incubated for 18 h at 37 °C. Subsequently, the cells were adjusted to McFarland 1.0, centrifuged at 5000× *g* for 10 min, and washed with sodium phosphate buffer (pH 7.0). Cell pellets obtained were suspended in fresh sodium phosphate buffer and were sonicated for 5 min on ice. The cell-free extract containing beta-lactamase enzyme was obtained by centrifuging the suspension at 16,000× *g* for 30 min and collecting the supernatant. Nitrocefin (a chromogenic cephalosporin) was used as the substrate to measure beta-lactamase enzyme activity. For this, 15 µL of the nitrocefin solution (0.5 mg/mL in 5% Dimethylsulfoxide) was mixed with 30 µL of the cell-free extract and the volume was adjusted to 100 µL using sodium phosphate buffer solution in a 96-well plate. The absorbance was measured using the plate reader in kinetic mode at 490 nm for 15 min.

### 3.2. H_2_S Production Inhibition Assay

The ability of test samples to inhibit the production of hydrogen sulfide (H_2_S) was measured as reported previously [30]. Bacterial isolates were adjusted to McFarland 0.5 and introduced to sub-lethal concentrations of samples and ampicillin (MIC × $\;\frac{1}{3}$) and incubated for 18 h in MHB medium with 20 mM L-cysteine. Bacterial cells were then collected by centrifugation at 5000× *g* for 10 min and washed twice with MHB + L-cysteine medium. The cells were adjusted to 0.5 McFarland and left to incubate for 30 min; 40 µL of the cell suspensions were added to 80 µL of the distilled water (pH 9.6 + 0.1 mM diethylenetriaminepentaacetic acid) in a microtiter well. Forty µL of the solution (17.1 mM *N*,*N*-dimethyl-*p*-phenylenediamine sulfate salt, and 37.0 mM FeCl_3_ in 6 M HCl) was added immediately. This method measures the H_2_S produced during bacterial growth in the broth. The absorbance was immediately detected in kinetic mode at 668 nm for 45 min using the plate-reader (Spectramax i3x, Molecular Devices, San Jose, CA, USA).

### 3.3. Efflux Pump Inhibition Assay

The ability of test samples to inhibit the efflux activity of the bacterial isolates was measured as mentioned previously with minor modifications [31]. The overnight culture of bacterial cells was washed and adjusted to McFarland 1.0. The cells were treated with sub-lethal concentrations (MIC × $\;\frac{1}{3})$ of the samples and CPZ, vortexed and incubated for 30 min. The sub-lethal concentration of EtBr (MIC × $\;\frac{1}{3})$ was added to the bacterial samples and 150 µL of the suspension was added to the individual wells of a 96-well plate. EtBr accumulation inside the bacterial cells was monitored (Appendix A) and followed by washing the cells with PBS and re-energizing with the addition of 10 µL of glucose (final concentration 0.04% *w/v)*. Efflux activity was then determined by monitoring fluorescence (530/590 nm) for 60 min using a plate reader.

### 3.4. Determination of Antibacterial Property of CMD against Intracellular S. enterica ser. Typhimurium

The antimicrobial effects of nanoparticle-encapsulated drugs against intracellular pathogens were determined as previously described [32]. Caco-2 cells (ATCC, Manassas, VA, USA) were selected as the in vitro model of the intestinal epithelium. Cells were cultured in a 96-well plate (2 × 10^4^ cells per well) until confluency and were infected with freshly cultured *S. enterica* ser. Typhimurium in MHB (Multiplicity of index = 1). After an hour of incubation, the supernatant media containing residual extracellular bacteria was removed and the cells were washed 2 times with PBS at 4 °C. The cells were then incubated with 100 µL of 200 µg/mL of gentamicin for 30 min to eliminate the extracellular bacteria. Gentamicin suspension was removed subsequently, and the cells were washed 3× with PBS at 4 °C. The cells were incubated with Gibco Dulbecco’s Modified Eagle Medium (DMEM) (Thermofisher, Tewksbury, MA, USA) for 4 h to establish the intracellular infection. Next, the NPs containing TET and/or CPZ were added to the cultures and incubated for 12 h. After 12 h, the cells were washed 3× with PBS at 4 °C to remove the extracellular bacteria. Subsequently, Caco-2 cells were lysed using 100 µL of 0.5% Triton-X and the lysate was serially diluted and plated on Mueller–Hinton agar plates. Infected cells with no treatment were used as negative control and infected cells treated with ciprofloxacin (20 µg/mL) were used as a positive control. The number of colonies was counted after incubation for 24 h at 37 °C to determine viable *S. enterica* ser. Typhimurium counts using the drop culture method [33].

### 3.5. Statistical Analysis

Experiments were performed in triplicates. The data are expressed as the mean value ± standard deviation (SD). *T*-test was performed and *p*-value ≤ 0.05 was considered significantly different.

## 4. Results

### 4.1. Antimicrobial Susceptibility

All the seven bacterial isolates were resistant to more than one antibiotic, as shown in Figure 1. Resistance to tetracycline and spectinomycin was observed in all the isolates. *S. enterica* ser. Typhimurium was the most resistant isolate, with a resistance to 75% of the tested antibiotics, followed by *S. enterica* ser. Dublin (58.3%). *S. hyicus* M48 resisted only 12.5% of tested antibiotics and was the most susceptible bacterial isolate. Results on the antibacterial resistance mechanisms such as efflux activity and beta-lactamase activity are summarized in Appendix A, where all *Salmonella* isolates showed efflux activity. Similarly, all except *S. hyicus* M48 of *Staphylococcus* isolates showed efflux activity. Beta-lactamase enzyme activity was shown by all isolates except for *S. hyicus* M48. In addition, biofilm formation ability and hemolysis were also determined as mentioned previously [34,35], and the results are summarized in Appendix A. *Staphylococcus* isolates mostly formed strong to moderate biofilm, while *Salmonella* isolates formed weak to no biofilm. *Staphylococcus* isolates caused complete hemolysis, except *S. hyicus* M48, which caused partial hemolysis. *Salmonella* isolates also caused hemolysis typical of HylE hemolysin protein.

### 4.2. Checkerboard Assay

Results of the checkerboard assay are summarized in Table 1. For *S. enterica* ser. Typhimurium, the ampicillin-tazobactam combination showed synergistic interaction (FICI-0.23) and tetracycline-cinnamon EO showed additive interaction (0.5 < FICI ≤ 1). Chloramphenicol-CPZ, chloramphenicol-thymol, and TET-CPZ combinations showed additive interaction against both *S. aureus* M12 and *S. enterica* ser. Typhimurium (0.5 < FICI ≤ 1). In *S. aureus* M12, synergistic interaction was observed with ampicillin-tazobactam (FICI-0.14), ampicillin-cinnamon EO (FICI-0.38), and chloramphenicol-oregano EO (FICI-0.5). Ampicillin-thymol, ampicillin-oregano EO, and tetracycline-oregano EO combinations were observed to have additive interactions against *S. aureus* M12.

Combinations of antibiotics (ampicillin, chloramphenicol, tetracycline, and trimethoprim) and adjuvants (efflux pump inhibitor chlorpromazine (CPZ), beta-lactamase inhibitor tazobactam (TAZ), essential oils—thymol (TML), cinnamon essential oil (CEO) and oregano essential oil (OEO)) were tested against *S. enterica* ser. Typhimurium and *S. aureus* M12 using the checkerboard assay; synergistic (SYN), additive (ADD), indifferent (IN).

### 4.3. Physico-Chemical Characterization of the Synthesized Particles

The hydrodynamic size and the zeta-potential of the synthesized particles were measured using DLS, as summarized in tabulated data (Figure 2). Zeta potential and the mean size of synthesized Ch NPs were observed to be +26.55 ± 4.73 mV and 536.30 ± 10.61 nm, respectively. On addition of TET to Ch NPs, the zeta potential and mean size did not change significantly and were +25.72 ± 0.68 mV and 541.68 ± 2.02 nm, respectively. The size of Si NPs encapsulating CPZ was determined to be 514.20 ± 53.05 nm, and the zeta potential changed significantly from −24.18 ± 0.08 mV to −9.95 ± 0.43 mV after the addition of CPZ. The size of Si NPs was observed to be 778.01 ± 8.16 nm before encapsulation mainly due to the presence of aggregation in Si NPs [36]. In CMD, the average particle size and zeta potential were observed to be 577.00 ± 35.83 nm, and +15.25 ± 0.11 mV, respectively. Through SEM imaging, the size of Si-CPZ NPs was found to be between 315.5 nm and 416.5 nm. On average, the size of Ch-TET NPs appeared to be 33.49–59.17 nm when observed under SEM, which, however, was found to be >500 nm when examined using DLS [36]. The Si-CPZ NP and Ch-TET NP display spherical shapes as evidenced by SEM examination. CMD, however, appeared to be a dumbbell-shaped cluster with individual particles around 457.1–650.7 nm (Appendix A).

The FTIR spectra (Appendix A) of Si NPs showed broad peaks in the region from 1000–1250 cm^−1^, which is attributed to Si-O vibrations [37]. The characteristic peaks of CPZ (1563 cm^−1^, 1457 cm^−1^, and 753 cm^−1^) were visible in Si-CPZ NPs, suggesting the presence of CPZ in the synthesized particle [38]. Peaks at 1637 cm^−1^ and 1540 cm^−1^ in Ch NPs shifted to 1601 cm^−1^ and 1524 cm^−1^, respectively, in Ch-TET NP. The shift in the IR spectrum peaks confirmed the formation of new bonds and encapsulation of TET into Ch NPs [11].

### 4.4. Encapsulation Efficiency and Release Behaviour

The EE was measured spectrophotometrically by measuring the absorbance of the supernatant at 285 nm for TET and 305 nm for CPZ. EE of TET in Ch NPs was determined to be 23.82 ± 0.16% and EE of CPZ in mesoporous Si NPs was 44.77 ± 0.12%. The release of TET from Ch NPs and of CPZ was measured by monitoring the absorbance of the supernatant over seven days (Figure 3). It was found that the release was faster for both TET and CPZ when the pH of suspending solution was 4.4. On day 1, 67.48 ± 1.57% of TET was released from Ch NPs and 81.49 ± 0.29% of CPZ was released from Si NPs at pH 4.4. At pH 6.3, 52.31 ± 0.12% of TET and 61.95 ± 0.91% of CPZ were released on day 1. Following the initial burst release, the release was relatively slower over the next six days.

### 4.5. Antibacterial Efficacy

The MIC of the synthesized particles and the CMD were determined against *S. enterica* ser. Typhimurium and *S. aureus* M12 and the results are summarized in Table 2. For *S. enterica* ser. Typhimurium, the MIC of Ch-TET NP was determined to be 1000 µg/mL, and the MIC of Si-CPZ NP was 500 µg/mL. MIC of CMD was 250 µg/mL, whereas the actual concentration of TET was only 29.77 ± 0.20 µg/mL. A decrease in MIC was also observed against *S. aureus* M12 when exposed to CMD in comparison with other free drugs or particles containing a single drug. Accordingly, the MIC of Ch-TET NP was 500 µg/mL, and that of Si-CPZ NP was measured to be 250 µg/mL. However, the MIC of CMD against *S. aureus* was 125 µg/mL, whereas the concentration of TET was 14.89 ± 0.10 µg/mL.

### 4.6. Beta-Lactamase Enzyme Inhibition Assay

Results from beta-lactamase enzyme activity in MDR bacteria treated with different test samples are given in Figure 4a. *S. aureus* M12 treated with CMD had significantly less (*p* < 0.05) beta-lactamase enzyme activity (6.98 ± 0.81 U/mL) than control group bacteria treated with ampicillin (22.58 ± 2.19 U/mL) and those treated with Ch-TET NP (14.73 ± 1.40 U/mL). While the inhibition was slightly lower for bacteria treated with Si-CPZ-NP (2.90 ± 1.30 U/mL), the values were not significantly different for CMD-treated bacteria, which suggested that CPZ in CMD is inhibiting beta-lactamase activity. A similar tendency was observed in *S. enterica* ser. Typhimurium, treated with CMD and other combinations. Ampicillin-treated *E. coli* ATCC 25922, and ampicillin-treated *S. aureus* ATCC 25923 were used as a negative control.

### 4.7. H_2_S Production Inhibition Assay

We tested the effect of drug-loaded particles on H_2_S production by bacteria. H_2_S protects bacteria from the damage induced by antibiotics, thus is an important mechanism in AMR bacteria [39]. The result of H_2_S production by bacterial isolates after treatment with particles is given in Figure 4b. It was observed that H_2_S production by *S. enterica* ser. Typhimurium was inhibited the most by CMD followed by Si-CPZ NP and Ch-TET NP. In the case of *S. aureus* M12, H_2_S production was inhibited the most by Ch-TET NP, followed by CMD. H_2_S production inhibition was lowest in Si-CPZ NP. *E. coli* ATCC 25922 and *S. aureus* ATCC 25923 were used as negative controls, where no H_2_S production was observed.

### 4.8. Efflux Pump Activity Inhibition

The differential ability of drug-loaded nanoparticles to block the efflux pumps, measured by the time taken for extrusion of EtBr in bacteria is given in Figure 4c. It was observed that Ch-TET NP treated bacterial cells of both *S. aureus* M12, and *S. enterica* ser. Typhimurium had $t_{2}^{1}$ > 3600 s for extrusion of EtBr, which is mainly due to extremely low accumulation of EtBr to begin with instead of efflux inhibition by nanoparticle treatments. Si-CPZ NP treatment led to the slowest extrusion of EtBr with $t_{2}^{1}$ of 728.5 s and 684.7 s for *S. enterica* ser. Typhimurium and *S. aureus* M12, respectively. CMD also successfully inhibited the efflux of EtBr with $t_{2}^{1}$ of 319.3 s and 584.2 s for *S. enterica* ser. Typhimurium and *S. aureus* M12, respectively. *S. aureus* ATCC 25923 and *E. coli* ATCC 25922 were used as negative controls.

### 4.9. Antibacterial Efficacy against Intracellular S. enterica ser. Typhimurium

Antibacterial efficacy of drug-encapsulated particles against intracellular *S. enterica* ser. Typhimurium in the Caco-2 cell model was tested. The colony-forming units (CFU) of the intracellular bacteria were determined (Appendix A) and log reduction for the treatments was calculated (Figure 5). It was determined that CMD (250 µg/mL) had a significantly higher log reduction (0.77 ± 0.26) than other treatments, which was an 83.02 ± 14.35% reduction. Log reduction for individual Ch-TET NP and Si-CPZ NP was lower and was only 0.01 ± 0.05 and 0.13±0.03, respectively, which is a 2.64 ± 11.21% and 25.85 ± 5.60% reduction. TET in suspension (not loaded in nanoparticle) had an only marginal reduction, with log 0.17 ± 0.13, whereas free CPZ showed no reduction in the bacterial colonies.

## 5. Discussion

Seven pathogen isolates containing *Salmonella* and *Staphylococcus* from animal farms were characterized to gain insight into the antibiotic resistance, mechanisms of resistance, and virulence factors for custom design of an antibacterial strategy. Based on the AMR and virulence characteristics, a nano-enabled combination therapy containing TET as an antibiotic and CPZ as an adjuvant was developed and tested successively against MDR *S. enterica* ser. Typhimurium and *S. aureus* M12.

The antimicrobial susceptibility profile showed that most of the pathogens were resistant to TET, ampicillin, lincomycin, and penicillin, which include some of the most widely used antibiotics in agriculture [40]. MDR bacteria use multiple mechanisms to overcome the action of antimicrobials, such as active efflux or enzymatic degradation of the antimicrobial, mutation of an antimicrobial target site, and reduced cell wall permeability [41,42,43]. In *Salmonella,* the most common efflux pump system, belonging to the RND family is the AcrAB-TolC, responsible for resistance to various antibiotics [44]. Likewise, we observed efflux activity in the tested *Salmonella* isolates, which might explain their strong resistance to several tested antibiotics (Appendix A). Similarly, the action of chromosome-mediated efflux pumps such as Nor and plasmid-mediated Qac efflux pumps [45] are reported to extrude biocides and antibiotics, explaining the efflux activity found in *Staphylococcus* isolates in our study (Appendix A).

Another major mechanism of resistance is the action of the beta-lactamase enzyme which hydrolyzes the beta-lactam ring. All tested bacterial isolates, except for *S. hyicus* M48 were positive for beta-lactamase enzyme activity. Beta-lactamase enzyme activity coded by *blaZ* genes has been reported in AMR *Staphylococcus* strains [44], while that in *Salmonella* has been associated with genes of *bla*_TEM,_
*bla*_CTX-M_, *bla*_SHV,_
*bla*_OXA-1_, and *bla*_CMY-2_ [44]. H_2_S production in bacteria is vital for protection against antibiotic-induced damage and oxidative stress. It is responsible for sequestering Fe^+2^ ions that can damage the bacterial DNA through the Fenton reaction. In addition, it stimulates the activities of antioxidant enzymes as well as superoxide dismutase. Reduction in H_2_S production is thus an indicator of subduing the general mechanism of resistance in bacteria [39]. Hemolysis and biofilm formation were investigated because they are two key virulence factors promoting pathogen survival in animals [46,47]. Tested *Staphylococcus* isolates were strong biofilm formers, while *Salmonella* isolates formed very weak or no biofilms (Appendix A). In addition, *Staphylococcus* isolates mostly manifested complete hemolysis (beta-hemolysis) [46], while *Salmonella* isolates showed hemolysis typical of HylE hemolysin protein (Appendix A), a pore-forming protein that contributes to the systemic pathogenesis of *Salmonella* [48]. The observation of the prevalence of multi-drug resistance and virulence characteristics in farm isolates highlights the need to develop alternate field-deployable antibacterial strategies. Combination therapy, wherein more than one antibiotic, or antibiotic with effector molecules are combined, has been deployed for the clinical management of infectious diseases in animals and humans [49,50].

We screened various combinations of antibiotics and effector molecules based on results from bacterial profiling for AMR and virulence characteristics. Synergism observed in the ampicillin-tazobactam combination could be ascribed to the action of tazobactam on the beta-lactamase enzyme, which was highly prevalent in both isolates [42]. Similarly, the synergistic interaction of the ampicillin-cinnamon EO combination could be attributed to membrane disruption by the EO, allowing easy passage of ampicillin into the cells. In addition, ampicillin and cinnamon EO share a common mechanism of action and work together to disrupt the bacterial cell membrane [43]. Yet another synergism was observed in the oregano EO-chloramphenicol combination against *S. aureus* M12. Oregano EO consists mainly of thymol and carvacrol, which affects the respiratory and energy metabolism in addition to membrane disruption [43]. This can damage the pH homeostasis and further lead to disruption of LmrS efflux pumps, which confer resistance to chloramphenicol, thus increasing the susceptibility to chloramphenicol [45]. CPZ interferes with substrate binding and thus inhibits AcrB-mediated efflux in *Salmonella* [51]. In *Staphylococcus*, CPZ impacts the potassium flux across the membrane, and thus disrupts the TetK efflux pumps, responsible for the extrusion of TET [52]. In addition, CPZ has shown to impede the transfer of plasmids between bacteria whereby curtailing the horizontal transfer of resistance genes [53]. Therefore, CPZ holds the ability to sensitize bacterial cells toward TET and thus increase its effectiveness, thus the combination had an additive effect as shown in Figure 6.

TET is classified as Category III antibiotics according to Health Canada and is not the preferred option for the treatment of serious human infections [4]. It is permitted for use in food animals, but due to its overuse, the frequency of AMR pathogens has increased lately [54]. The combination of TET and CPZ was chosen for nanoencapsulation because, (1) TET is a commonly used antibiotic in animal agriculture that needs improvement in performance against the rising AMR, and (2) this combination showed an additive effect (not synergistic), which is more appropriate to demonstrate the advantage of encapsulating drugs in nanoparticles.

Nanomaterials of silica and chitosan were chosen as drug carriers in this study as their biocompatibility profiles are well established and are used as carriers of bioactive molecules in drug formulations [55,56]. It should be noted that CMD is designed to be administered through the oral route to treat the gastrointestinal tract (GIT) and not through the intravascular route. Unlike the intravascular route of administration that may trigger thrombophlebitis (an inflammatory process that causes a blood clot), oral drug administration is less invasive and thus safer [57].

The combination of Ch-TET NPs and Si-CPZ NPs (CMD) showed an improved antibacterial effect and successfully inhibited the H_2_S production, efflux activity, and beta-lactamase enzyme activity in treated bacterial cells. The MIC value of CMD was 250 µg/mL and 125 µg/mL for *S. enterica* ser. Typhimurium and *S. aureus* M12, respectively. It should be noted that the effective concentration of TET and CPZ was only 23.82 ± 0.16% and 44.77 ± 12% in Ch-TET NPs and Si-CPZ NPs, respectively. As such, there was an 88.09% and 76.18% reduction in the MIC values of TET against *S. enterica* ser. Typhimurium and *S. aureus* M12, respectively, when it was present in CMD.

The enhanced antibacterial efficacy of the combinations of drugs encapsulated into nanoparticles can be attributed to two main factors: (1) the additive effect between TET and CPZ, as observed from checkerboard assay result, and (2) the improved localized delivery of drugs to bacterial cells. Apart from its inhibitory action on efflux activity, Si-CPZ, NP, and CMD were noted to significantly lower the production of the beta-lactamase enzyme activity in both *S. enterica* ser. Typhimurium and *S. aureus* M12. This observation is in line with the reported inhibitory action of CPZ on the expression of *bla* and *mecA* genes [58]. The combined effect shown in suspension culture may not be achieved if free drugs are administered to the human or animal body. This is because the therapeutic efficacy of systemically or orally administered free antibiotics can be hindered due to various obstacles: limited bioavailability, poor target specificity, faster degradation, and body clearance [59]. Nanoparticle-based drug delivery systems offer the opportunity of enhancing the efficacy of antibiotics while minimizing the side effects [60]. Further, the ability of CMD to significantly reduce the number of intracellular MDR *S. enterica* ser. Typhimurium points towards the superior properties of nanoparticle drug carriers to target bacteria taking refuge inside host cells. CMD had a positive surface charge, which is beneficial for oral administration of drugs as the positive charge of CMD could penetrate the mucous layer on the intestinal epithelial cells, the growth site of bacteria causing gastrointestinal diseases in animals [61]. In short, the superior performance of CMD stems from the additive action of antibiotic (TET) and effector molecule (CPZ) and delivery of effective concentrations of drugs to the bacteria-laden cells achieved through nanoparticles.

The ability of CMD to reduce intracellular pathogens also opens new pathways for preserving and enhancing the antibacterial efficacy of various commonly used antibiotics in food animals, to which various pathogens have developed resistance. Thus, the nano-enabled combination therapy extends the usability of category III antibiotics in eliminating tougher-to-treat infections, thereby slowing down the need to rely on last-resort antibiotics for treating animal infections. While this effect was exemplified by combinations of TET and CPZ, we believe similar strategies could extend the usability and effectiveness of many other antibiotics permitted for use in food animals.

## Figures and Tables

**Figure 1 nanomaterials-12-02179-f001:**
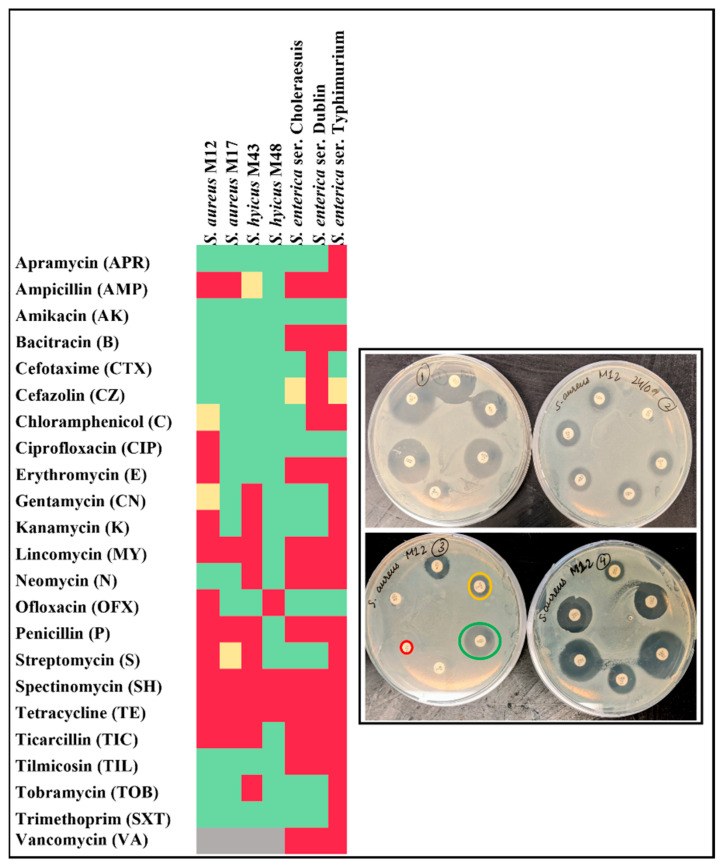
Antimicrobial susceptibility of bacterial isolates: Antimicrobial susceptibility of bacterial isolates was tested using the Kirby–Bauer disk diffusion assay. Disks containing antibiotics were placed on the lawn of bacteria and incubated overnight at 37 °C (as seen in the inlet). The zone of inhibition was measured, and isolates were categorized as resistant (red), intermediate (yellow), susceptible (green), or not applicable (grey) according to CLSI guidelines.

**Figure 2 nanomaterials-12-02179-f002:**
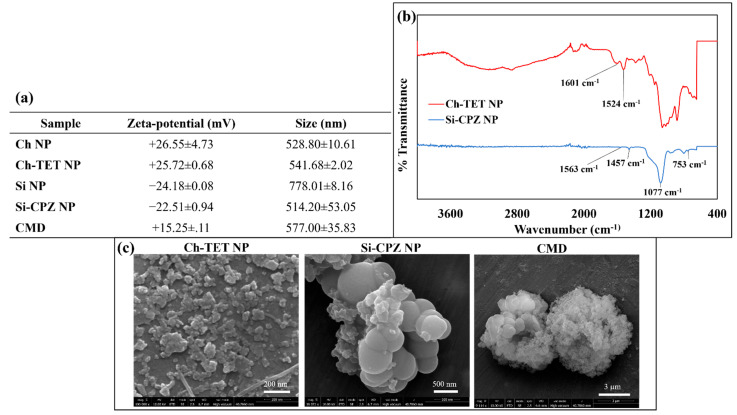
Characterization of particles. (**a**) Tabulated data showing zeta-potential, and the size of the particles measured using dynamic light scattering (DLS). (**b**) FTIR spectrum of Ch-TET NP and Si-CPZ NP. (**c**) Nanoparticles were platinum-coated and imaged using Scanning Electron Microscopy (SEM).

**Figure 3 nanomaterials-12-02179-f003:**
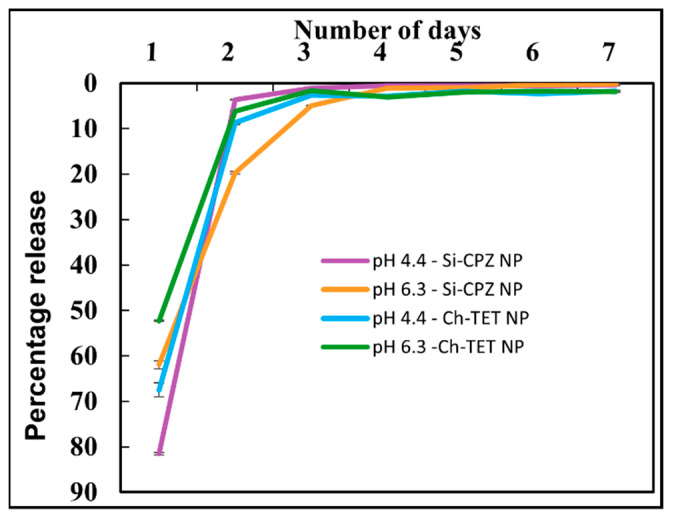
Release kinetics of drugs from nanoparticles. Release kinetics of tetracycline (λ = 285 nm) and chlorpromazine (λ = 305 nm) from chitosan nanoparticles and silica nanoparticles were measured, respectively, using a spectrophotometer.

**Figure 4 nanomaterials-12-02179-f004:**
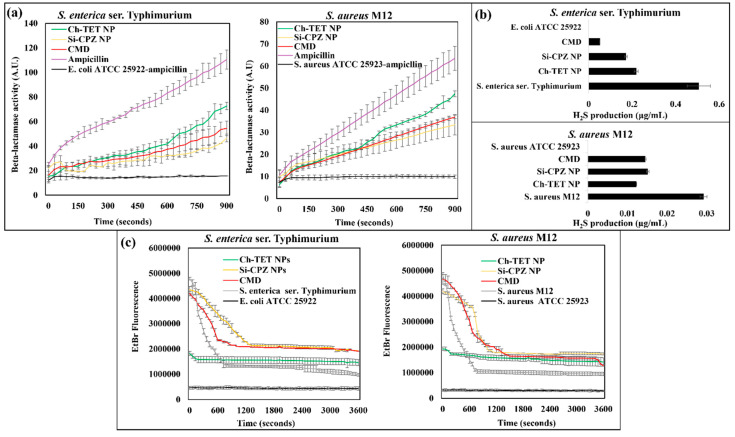
Effect of synthesized nanoparticles on resistance mechanisms. (**a**) Inhibition of beta-lactamase enzyme in bacteria treated with synthesized nanoparticles measured using nitrocefin assay, (**b**) H_2_S production of bacterial strains was measured after treatment with synthesized particles, and (**c**) EtBr extrusion (efflux) from bacterial cells after treatment with nanoparticle treatments was determined by measuring fluorescence.

**Figure 5 nanomaterials-12-02179-f005:**
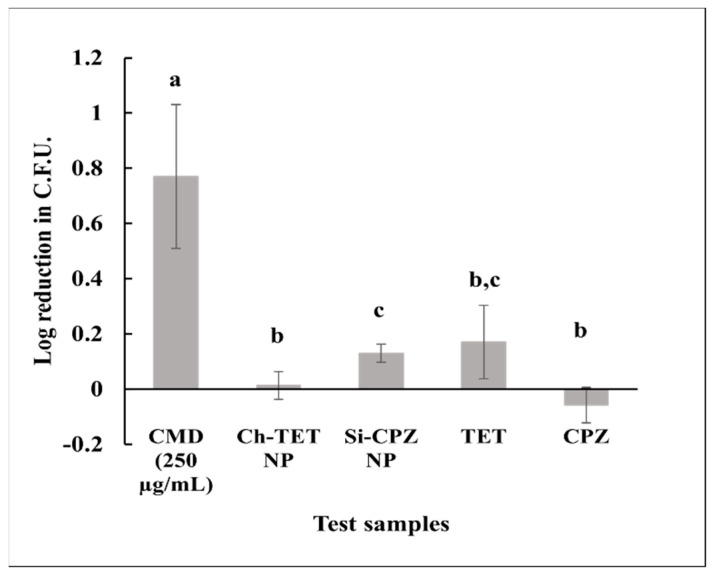
Effect of nanoparticle treatments on intracellular bacteria. Infected Caco-2 cells were treated with nanoparticle samples for 24 h. The cells were lysed, and the lysate was used to count the intracellular bacteria that survived. Log reduction was calculated. *T*-test was performed and *p* < 0.05 was considered significantly different.

**Figure 6 nanomaterials-12-02179-f006:**
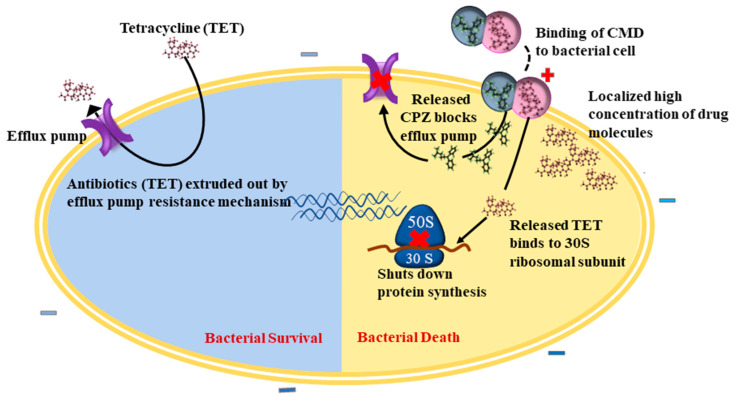
Mechanism of action of CMD on the bacterial cell. Positively charged CMD binds to negatively charged bacterial cells and releases TET and CPZ. CPZ blocks the efflux pumps and inhibits the extrusion of TET. TET binds to the 30S ribosomal subunit and shuts down the protein synthesis, leading to cell death.

**Table 1 nanomaterials-12-02179-t001:** Antibacterial effect of combinations of antibiotics and adjuvants.

	*S. enterica* ser. Typhimurium					*S. aureus* M12				
	CPZ	TAZ	TML	CEO	OEO	CPZ	TAZ	TML	CEO	OEO
Ampicillin	IN	SYN	IN	IN	IN	IN	SYN	ADD	SYN	ADD
Chloramphenicol	ADD	IN	ADD	IN	IN	ADD	IN	ADD	IN	SYN
Tetracycline	ADD	IN	IN	ADD	IN	ADD	IN	IN	IN	ADD
Trimethoprim	IN	IN	IN	IN	IN	IN	IN	IN	IN	IN

**Table 2 nanomaterials-12-02179-t002:** Minimum inhibitory concentration (µg/mL) of Ch-TET NP, Si-CPZ NP and CMD.

	Si-CPZ NP (µg/mL)	Ch-TET NP (µg/mL)	CMD (µg/mL)
*E. coli* ATCC 25922	62.5	7.82	3.90
*S. aureus* ATCC 25923	125	3.90	1.95
*S. enterica* ser. Typhimurium	500	1000	250
*S. aureus* M12	250	500	125

## Data Availability

Data available upon request.

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
