# Peer review of "Nanoparticle-Enabled Combination Therapy Showed Superior Activity against Multi-Drug Resistant Bacterial Pathogens in Comparison to Free Drugs"

_nanomaterials, 2022, doi:10.3390/nano12132179_

Round 1
Reviewer 1 Report
The authors present a comprehensive study on the potentisation of antimicrobials and nanoparticle encapsulation. They did a lot of work but, unfortunately, not all that work was necessary. First of all, I am concerned with the choice of compound for tests. According to Wang et al. (Nature 2019, doi.org/10.1038/s41396-018-0275-x), carbamazepine promotes the transfer of AMR genes. Therefore, in my opinion, the use of its combination with antimicrobials should be avoided. Second, some parts of the experiment are unnecessary. For example, the authors test the formation of biofilm but then they do not verify the action of the tested compounds on biofilm. They test combinations of different compounds on bacteria, including those with obvious synergy, like ampicillin + tazobactam. Finally, the design of some experiments is flawed. In the lactamase assay, it is rather AMP that induces the expression of the enzyme, not CMD that inhibits it. Why was AMP not added to all groups? Is there a significant difference between TET NP and CMD?
The paper would benefit if AMR mechanisms were in-depth described. These are just two isolates so WGS could be performed. Then in the discussion (focusing a lot on ARGs) you could relate to that.
Please describe the categories of antimicrobials to non-Canadian readers.
Reviewer 2 Report
1) Along with synthesis, authors should explain the properties of both silica and chitosan nanoparticles
2) Structural and Functional Properties like size, shape and porosity, surface modification of silica and chitosan nanoparticles
3) Biocompatibility and Biodistribution of both silica and chitosan nanoparticles
4) Author should provide the schematic diagram of the mechanism of action
Round 2
Reviewer 2 Report
Accept in present form